# *Mycobacterium tuberculosis*: Implications of Ageing on Infection and Maintaining Protection in the Elderly

**DOI:** 10.3390/vaccines10111892

**Published:** 2022-11-09

**Authors:** Victor Bonavida, Mitchell Frame, Kevin H. Nguyen, Shlok Rajurkar, Vishwanath Venketaraman

**Affiliations:** 1Department of Basic Medical Sciences, College of Osteopathic Medicine of the Pacific, Western University of Health Sciences, Pomona, CA 91766, USA; 2Division of Biological Sciences, University California Berkeley, Berkeley, CA 94720, USA

**Keywords:** tuberculosis, ageing, infection, mycobacterium tuberculosis, immune, mTOR, autophagy, mitophagy, susceptibility

## Abstract

Several reports have suggested that ageing negatively affects the human body resulting in the alteration of various parameters important for sufficient immune health. Although, the breakdown of innate and adaptive immunity has been hypothesized to increase an individual’s susceptibility to infections including *Mycobacterium tuberculosis* (*M. tb*), little research has been done to bridge this gap and understand the pathophysiology underlying how ageing increases the pathogenesis of *M. tb* infection. Our objective was to study research from a plethora of resources to better understand the pathogenesis of ageing and its link to the human immune system. To achieve this goal, this article explores how ageing decreases the collective T-cell immune response, reduces glutathione (GSH) production, over activates the mammalian target of rapamycin (mTORC1) pathway, inhibits autophagy and mitophagy, and alters various protective genes/transcription factors. Specifically highlighting how each of these pathways cripple an individual’s immune system and increases their susceptibility from *M. tb* infection. Furthermore, research summarized in this article gives rise to an additional mechanism of susceptibility to *M. tb* infection which includes a potential defect in antigen presenting by dendritic cells rather than the T-cells response. Inflammaging has also been shown to play a role in the ageing of the immune system and can also potentially be a driving factor for increased susceptibility to *M. tb* infection in the elderly. In addition, this article features possible preventative strategies that could decrease infections like *M. tb* in this population. These strategies would need to be further explored and range from immunomodulators, like Everolimus to antioxidant supplementation through GSH intake. We have also proposed the need to research these therapies in conjunction with the administration of the BCG vaccine, especially in endemic populations, to better understand the risk contracting *M. tb* infection as well as ways to prevent infection in the first place.

## 1. Introduction

The global population is ageing through a combination of, among other factors, improved medical and hygiene practices, lower fertility, and the ageing of the “Baby Boomer” generation, born between 1946 and 1964. The percentage of people above the age of 60, currently at 11%, is expected to increase to 22% by 2050. This ageing is especially noticeable in developed countries and has been associated with an increase in the incidence of chronic health problems [1]. This trend of an ageing population provides a greater reason to investigate treatment options for conditions that become more common and severe with ageing, such as cancer, heart disease, diabetes, and tuberculosis [2,3,4]. The incidence and severity of such diseases are often associated with changes in biochemical pathways important for sustaining life and other physiological changes [5]. For instance, ageing is associated with a decrease in insulin sensitivity, increased risk of apoptosis, and various changes in mitochondrial and cardiovascular system function which contribute to the risk of developing conditions such as Type 2 Diabetes Mellitus and cardiovascular disease [5].

Ageing should also be seen as a microscopic process that can have numerous negative consequences for individual cells that can impact the body as a whole. Some of these consequences involve the normal functioning of the cell. Ageing has been linked with an increase in reactive oxygen species (ROS) in the mitochondria, which in turn has been linked with oxidative damage to cellular components, especially those involved with DNA and mRNA [6]. A cell’s mitochondrial membrane will also become decayed and more permeable with age, leading to the potential leaking of ROS and enzymes harmful to the cell outside of the mitochondria and into the cytosol, possibly stimulating apoptosis [7]. The ageing of a cell has also been linked with a shift from aerobic to anaerobic respiration, being much less efficient in ATP production and creating harmful waste products such as lactic acid. Lactic acid as a waste product from anaerobic respiration also contributes to lactic acidosis, which has negative consequences in terms of survival and overall health [8,9]. In general, mitochondrial dysfunction combined with the increased sensitivity to apoptosis associated with ageing, leads to increased risk of cellular degeneration and death, likely resulting in negative consequences from an infection resistance standpoint [5].

Ageing is also associated with decreased glutathione (GSH) production. GSH is an antioxidant present in the body that plays a vital role in the detoxification of harmful chemicals and the support of a healthy immune response. GSH reduces free radicals and ROS to form glutathione disulfide (GSSG), which is then reduced back into GSH. The reduction of these species prevents excessive oxidative stress within the cell, a factor in preventing diseases such as cancer and liver disease. GSH also regulates processes such as gene expression, apoptosis, and insulin sensitivity [10].

Importantly, ageing has detrimental effects on the immune system, T-cell exhaustion, and T-cell anergy, which involve the programmed death of T-cells, and the ineffectiveness of T-cells, respectively are commonly observed in ageing populations. A third deleterious process observed in the ageing population is immune senescence, where the shortening of telomeres caused by cellular division hinders their function and ability to divide [11,12,13,14].

## 2. Inflammaging

Within the last 20 years, a novel approach to immune system function, chronic diseases, infection susceptibility and other proinflammatory processes was conceptualized and coined “inflammaging” [15]. This new concept has carved new ways to consider the beneficial or detrimental complexities of ageing, age-related changes, and other age-related processes. As a result, we have now a new starting point in the approach for the ageing process in a well-rounded and holistic way. There have been many discoveries in how our immune system has evolved over time and been affected both in the innate and adaptive arms of the immune system [15].

The adaptive immune response is the most robust and complex in addition to allowing for memory-based immunity. One of the major changes of the adaptive immune system is the involution of the thymus. This involution leads to a decrease in naïve T-cells with a simultaneous relative increase in established amount of memory T-cells. This simple idea of less production of new T-cells in the thymus, while maintaining our levels of prior memory T-cells can explain as to why infection and cancers increase in the elderly and not responding to vaccines as well [15]. In summary, having less naïve T-cells leads to decreased recognition of new antigens from cancers or infections. Other changes seen in T-cells are a result of chronic stress or induction, leading to senescence and exhaustion. More of these exhausted T-cells increase their production of pro-inflammatory cytokines and these cytokines act on other immunological cells changing their gene expression and their ability to function [15].

The other arm of the immune system is the innate immune system. This side of the immune system also undergoes changes mostly affecting phagocytic cell activity and their function. While the way the innate system works, relative to the adaptive system, is much less complex, it still has its own intricacies and concepts to consider. Many phenotypical expressional changes occur in these cells as they are repeatedly activated throughout life. For example, macrophages become more inflammatory and act like exhausted or senescent cells, or natural killer (NK) cells shift towards a more cytotoxic phenotype. Many studies have shown that the functions of the innate immune system (phagocytosis, chemotaxis, killing, antigen presenting) all become altered with time and life progression [15].

Inflammaging has evolved since it’s conception over 20 years ago, but overarchingly the idea that a combination of genetics, epigenetics, and constant stressors as we age effect various systems, pathways, and cells of the immune systems in ways that lead to in inflammatory exhaustion or senescence. Once the immune system reaches this point many molecules and different modulators become activated pushing the system as whole towards a threshold. This threshold varies physiologically in everyone, but if that threshold is surpassed it leads down a road towards pathological aging. Hence, it is logical to reason by preventing the stressors in the first place, understanding the genetics involved can help better understand and prevent the adverse effects of inflammaging [15].

With this understanding of inflammaging and its role on disease states as we age, we can talk about its specific role on *M. tb* infection. Studies have shown that older mice have increased amounts of macrophages in an inflammatory state and levels of proinflammatory cytokine signals. With the increased levels of proinflammatory signals acting on the high number of inflammatory macrophages causes a viscous cycles leading to a net gain of inflammation within the lungs of these elderly mice. When infected with *M. tb* when compared to younger mice, the older mice had increased levels of proinflammatory cytokines secreted and increased uptake of the bacteria [16]. One other study showed changes in monocytic phenotypes and proportions with aging and *M. tb* infection, leading the researchers to believe that these changes can play a role in susceptibility in the elderly to contract active *M. tb*. This suggests that similar inflammaging changes were seen in these individuals and that low-grade chronic stressors can potentially increase susceptibility to *M. tb* infection in the elderly [17].

## 3. T-Cell Immune System

Aside from natural ageing, the immune system undergoes many types of other processes leading to altered function. These definitions should be clearly differentiated from one another [11]. The first process is described as T-Cell exhaustion. T-Cell exhaustion has been reported in various patients with infections including human immunodeficiency virus (HIV), hepatitis B virus (HBV), hepatitis C virus (HCV) infected individuals as well as various patients with cancers [11]. Exhaustion itself, has mostly been described as a CD8^+^ process that is hypothesized to take place due to multiple biochemical pathways [11]. Of these pathways, the programmed death-1 (PD-1) pathway seems to play a crucial part in T-Cell exhaustion [12]. During chronic infection, genetic expression in CD8^+^ T-cells show upregulated PD-1 receptors in chronically infected mice [12]. The upregulation of PD-1 leads to a downstream inhibition of the CD28 family within the CD8^+^ T-cells [13]. As a result of this, it was hypothesized that blocking PD-1 action may improve CD8^+^ T-cell effector function [13]. In patients with HIV-1 infection, high PD-1 correlated with increased viral load [13]. HIV infected mice were administered a PD-L1 monoclonal antibody, which decreased viral load, increased T-cell numbers, and activity as well as increased percentage of memory T-cell numbers [13]. Overall, T-cell exhaustion can play a role in one’s ability to fight and resolve infection.

The second pathway is described as anergy. As opposed to T-cell exhaustion, where the T-cell loses its effector function, T-cell anergy has been described as a process due to a low stimulatory and high inhibitory environment which leads to an induced hyporesponsive state of the T-cells [14]. In various cancers, there has been evidence of an imbalance between stimulatory and inhibitory factors within the T-cell microenvironment [14]. Tumors seem to display numerous B7 family proteins which leads to poor costimulatory interactions leading to increased anergic states of cells [14].

Additionally, there is immune senescence. Senescent T-cells are defined as those due to shortening of telomeres, phenotype changes and cell cycle arrest [14]. Shortening of telomeres happens due to an intrinsic process of cell division ultimately affecting cell function and DNA structure [14]. Cell cycle proteins, like p16, p21, which are normally responsible for inhibiting cell cycling progression, have been shown to be increased within senescent cells [14]. Lastly, senescent T-cells lose their killing abilities, and they function to be negatively inhibited and regulated [14]. With ageing, all three of these processes have been shown to be altered in some way. As we age, T-cell mitochondrial function has been shown to become dysregulated leading to increased senescence [18]. T-cells also seem to enter an anergic state more easily due to ageing effect on CD28 expression, rendering T-cells nonresponsive to certain antigens [19]. Finally, as we age, our T-cells spend more time fighting chronic infection and inflammation and it has been shown that over time, T-cells becomes exhausted due to this increased antigen stimulation [20]. Not only are each of these three processes; CD8^+^ T cell exhaustion, anergy, and senescence, crucially affected by ageing, but also by chronic infections such as *M. tb*. It has been found that cells in individuals who have been infected with *M. tb* have been discovered to have aged prematurely, leading to senescent cells, directly correlating with an *M. tb* survivor’s high risk of mortality [21]. Additionally, anergy in patients with active *M. tb* infections has been thought to be a response to monocytes actively producing cytokines that effectively suppress lymphocytes and promote fibrosis such as transforming growth factor β (TGF-β) [22]. Interestingly, a study has shown that CD8^+^ T cells, with a lack of co-expression of receptors, in individuals with *M. tb* has not shown effects of CD8^+^ T cell exhaustion [23] and therefore begs the question if ageing is the only factor affecting CD8^+^ T cell exhaustion? More research is needed to fully understand this connection. Figure 1 below illustrates these three processes of T-cell dysregulation seen in ageing populations.

## 4. *M. tb* and Granuloma Formation

Granuloma formation is a process that provides protection to the host by preventing replication and spread of *M. tb* infection. However, in the last decade a lot of research has been done showing that this process and its role is not as clear cut and has many nuances to it. The reason for this is the conflicting presentation of the type of role granuloma formation plays in infection. On one side, the process seems to provide a vessel for bacterial expansion and replication while on the other side this process seems to allow for sequestration and protection of bacterial products [24].

There is some research that shows that pathogenic forms of *M. tb* can hijack and manipulate host immune responses to take leukocyte and cell death processes to allow for bacterial replication and expansion. ESX-1, a bacterial secretion system is a gene that has been shown to play a role in the virulence as shown in vitro of human cells infected with *M. tb.* It was discovered that the region of difference 1 (RD1) locus contains the sequences for ESX-1 genes. It was shown that changes or mutations in RD1 showed a defect primarily in granuloma formation and showed decreased bacterial numbers suggesting that granuloma’s role points more towards expansion of bacteria rather than protection when compared to wild-type *M. tb* strains [24].

On the other hand, other researchers have suggested that while *M. tb* may be able to induce and use granuloma formation for its own expansion, others say that the granuloma formation can serve protective functions as well. While these mechanisms are not fully understood and hard to decipher, there are some points made that would indicate this can be true. For example, granulomas contain T-cells within them and help transfer protective immunity to within the walls of the granuloma. Hence, inherently there are protective cells found within the granuloma itself that we know play a critical role in inhibiting *M. tb* infection. Furthermore, in those populations, like the elderly, with decreased T-cell immune responses, macrophages cannot be properly activated by T-cells. Another example is one shown in a study of HIV infected individuals who had concomitant *M. tb* infections. HIV individuals have decreased granulomatous responses and in turn decreased control of *M. tb* infection [25].

Overall, it is possible that mycobacteria might benefit from the granuloma formation process and whether there is any protective function has truly been left unproven. However, there is the data supporting that the formation of them do indeed provide a bacterially favorable environment to allow for expansion. Hence, when looking at possible treatment options, especially in elderly populations, slowing of granuloma formation should be considered [24]. At the same time, the inherent possible protective functions of granuloma formation and action also should be considered and studied further [25].

## 5. Immune Response of *Mycobacterium tuberculosis* in Young Healthy Individuals

### 5.1. Initial Immune Responses

The human immune system can be broken down into the innate and adaptive responses. Both systems are utilized in defending ourselves against *M. tb* [26]. *M. tb* is transmitted through inhalation of droplets that have bacteria in them bringing the organisms into the lungs [26]. The innate immune response plays a critical role in the initial exposure to *M. tb* as well as the control and progression of possible long-haul infection. This is accomplished by constantly reeducating and stimulating adaptive immune cells regulating the inflammatory response [26]. Initially, *M. tb* pathogen associated molecular patterns (PAMPs) are recognized by a whole host of various receptors that induce opsonic and non-opsonic uptake of the bacteria [26]. Mycobacterial components are recognized by pattern recognition receptors which induce cytokine release and autophagy [26]. Macrophages are the first cell to encounter the bacteria during infection and the recognition leads to phagocytosis to sequester and hopefully eradicate the pathogen [26].

Macrophages attempt to contain the infection through lysosomal fusion and eventual acidification within the phagolysosome [26]. One of the major host receptors that interacts with the *M. tb* components are Toll-like receptors (TLRs). Mutations and polymorphisms in TLR genes have been shown to increase susceptibility to *M. tb* infection [27]. The interaction between bacterial components acting as ligands and TLRs leads to many downstream effects and signals. Ultimately, leading to the activation of NF-κB pathway, upregulating nuclear transcription of immune and inflammatory mediators [27]. IFN-γ, one of the major inflammatory signals, is a T-cell derived cytokine that induces macrophages to become activated and enter their killing state [28]. TNF-α is produced from macrophages and dendritic cells and has been shown to be one of the most important cytokines in controlling *M. tb* infection [28].

Reactive oxygen species (ROS) are another molecule crucial in infection control of phagocytosed pathogens [29]. Much of this ROS production occurs via the NADPH oxidase pathway via creation of superoxide thereby producing other free radical molecules like hypochlorite and hydrogen peroxide [29]. Patients with depleted NADPH oxidase systems were shown to have impaired control of *M. tb* infection [29].

Additionally, IL-1 and its family members of IL-1α and IL-1β have been shown to be increased during early *M. tb* infection [29]. IL-1β is a unique cytokine whose secretion is independent of ER-Golgi pathway and is formed as a pro-peptide that gets cleaved by a caspase to be activated [28]. This pathway is complex and controversial but still is understood to be one of the most prominent proinflammatory cytokines assisting in protection from tuberculosis [28].

Autophagy is described as cytoplasmic materials being degraded and/or recycled and is prevalent in *M. tb* infections [29]. Genes in relation to autophagy pathways have been shown to be involved in reduction of intracellular bacterial load [29]. There is evidence to support that autophagy genes are integrated into the host genome response to infection via numerous pathways [26]. *M. tb* leads to damage of phagosomes leading to downstream ubiquitin activated autophagic signaling eventually leading to autophagy of the infected cell [29]. Ultimately, these molecules, mediators, and signals lead to regulatory functions of autophagy pathways and increased intracellular killing of *M. tb* and impaired bacterial replication [26]. Following these initial defenses, most dominant in alveolar macrophages within the lung, chemokines can be secreted to increase host effector mechanisms against *M. tb* infection [26].

Overall, these pathways are complex and numerous, while surface level ideas are touched on in this review, more details of the immune response can be found in Barber, 2015 et al. [30].

### 5.2. Latent M. tb

Latent *M. tb* infection (LTBI) is largely misunderstood as there are many gaps of knowledge when it comes to host defenses in this type of infection. However, there are a few concepts we know are critical to the host and its ability to suppress the active infection. First and foremost, T-cells play an important role as seen from experimental findings resulting in HIV patients with depleted CD4^+^ cell counts have increased *M. tb* risk. There have also been experiments showing that when MHC-II is knocked out of CD4^+^ cells there is also an increased risk [31]. They way CD4^+^ mount this type of defense and the antigens they bind to are not well understood. CD4^+^ cells also seem to play a role as an effector on other immune cells within the response. These CD4^+^ cells increase antibody formation from B cells as well as increase CD8^+^ T cell activity. Recent data has also shown that CD8^+^ cells specifically play a prominent part in preventing spread of *M. Tb* throughout the body. While we know some of this information for certain, many things remain unknown. According to Boom et al, we have yet to discover numerous things like which antigens on infected cells are recognized by T-cells, Is *M. tb* can decrease antigen processing limiting the number of antigens presented to T-cells? What is the actual role of CD8^+^ cells in protection against LTBI? These among other questions have yet to be answered and probe for further research [31].

While protection is an important factor behind keeping *M. tb* in a latent state, we must also try to understand how *M. tb* escapes or avoid the latent stage to begin with. *M. tb* has capabilities of inhibiting MHC-II antigen processing, inhibition of the T-cell receptor and CD3 signaling, and antigen escape mechanisms. It is these pathways as a singleton or as a collective that allow for the bacteria to evade host immune systems and remain active within the host [31]. Again, while we understand these few mechanisms, many things remain to be gaps in our knowledge. For example, do these evasion tactics effect other cells other than CD4^+^ T-cells? Which eversion pathway is utilized the most? When do the infection and disease states effect the immune system in way to be able to evade it in the first place [31]?

In summary, many things are misunderstood when it comes to latent infection as well as the contributing factors to active versus latent disease states. However, we do know of the many cell types that do play a role, various cytokines involved in signaling and the ways *M. tb* can evade the host immune system which all are leading us closer to fully understanding the immunological response to this prevalent and deadly infection. The important mediates of host defense to *M. tb* is summarized below in Table 1.

### 5.3. GSH System

Glutathione (GSH) is the most abundant thiol present in cells which plays an important role in host antioxidant defense [10]. Along with oxidative stress, it also plays a role in nutrient metabolism and regulation of cellular processes, like gene expression, protein synthesis, cell proliferation, cell apoptosis, cytokine signaling and more [10]. Deficiency in GSH contributes to increased oxidative stress contributing to various diseases and ageing [10]. GSH is made de novo in most mammalian cells via a two-step enzymatic process [32]. The first step utilizes cysteine and glutamine by a ligase enzyme converting them into γ-glutamylcysteine, acting as the rate limiting step in GSH synthesis [32]. After one other enzymatic step, GSH is formed which can perform its antioxidant activity [32]. The mechanism by which GSH can execute its function as an antioxidant is through a series of redox reactions [32]. GSH peroxidase takes 2 GSH molecules as a substrate and links them together while reducing hydrogen peroxide into water [32]. The linking of these 2 GSH molecules allows for a disulfide bridge to be formed turning GSH into the oxidized form of glutathione, glutathione disulfide (GSSG) [32]. The GSSG form is no longer able to perform oxidation protection, but GSH can be reformed via glutathione reductase [32]. This enzyme takes the GSSG molecule and reduces it back to its original form of GSH using NADPH as a cofactor [32]. This is turn converts NADPH into an NADP^+^ molecule [32]. It is this recycling and reusing system that allows for cells to battle the oxidative stress within cells, however, this process can become saturated and lose its ability to clear the ROS [32]. As discussed previously, ROS and nitric oxide species play a role in infection defense, especially in *M. tb* [32]. As these products are made, GSH will be synthesized and will undergo continual redox reactions to help suppress the oxidative damage potential [32]. Many of these ROS include superoxide, hydrogen peroxide, hydroxyl free radicals and singlet oxygen which can all contribute to infection control [32].

Nitric Oxide is another major molecule in the control of *M. tb* infection [32]. It could interact with GSH to form S-nitrosoglutathione (GSNO), which can carry NO and release it leading to toxicity and death of organisms [32]. There have been a few reasons proposed as to how and why GSH can have antimycobacterial effects [32]. First, mycobacteria cannot make GSH, which leads to an imbalance in the redox ability within the bacterium when exposed to increased levels of GSH and inhibits bacterial growth [32]. This is due to mycobacterium using mycothiols as their molecule in regulating their redox activity [32]. The second reason being hypothesized is that GSH has been thought to be a sort of intrinsic antibiotic in advanced eukaryotic organisms [32]. GSH shows similarities to other antibiotics like those made from Penicillins and Cephalosporins, which may hint at the idea of GSH acting as a universal type of evolutionary antibiotic that may damage mycobacterium due to sensitivity to this molecule [32]. Lastly, with GSH acting as a precursor to GSNO, it has increased antimycobacterial effects by improving the response of macrophages and altering the cytokine profile within mycobacterium and infected cells [32].

### 5.4. Autophagy and Mitophagy

Autophagy is a form of degradation mediated via lysosomal pathways that plays an important regulatory role in cellular, tissue and organism homeostasis [33]. Autophagy-related genes (ATG) control the various types of autophagy present in cells, which have been categorized as chaperone-mediated, microautophagy and macroautophagy [33]. ATG gene’s function was discovered to be responsible for forming double membranes to deliver contents to lysosomes for degradation [33]. This process is conserved to be seen in almost all eukaryotic cells impressed upon both intracellular and extracellular signals [33]. In a general sense, once signals have been activated, the process can begin as cytoplasmic contents can be organized and group together to form a phagosome structure to become fused with a lysosome [33]. From there autophagy can regulate a complex and long list of functions like pathogen elimination, dead cell clearance, autoimmunity protection, adaptation to genomic instability, reactive oxygen stress, organelle quality, inflammatory signaling and many more [33]. Ultimately, autophagy leads to increased cell survival by turning over the dysfunctional cell products preserving its homeostatic state [33].

In *M. tb* infection, increased autophagy activity has been shown to suppress the activity of the pathogen intracellularly [34]. There is also increasing evidence that autophagy contributes to the containment and killing of intracellular invading pathogens including *M. tb* [34]. Xenophagy is the process by which bacteria can be eliminated by degradation and was first discovered in relation to *M. tb* [35]. It was shown that stimulation of autophagy within host macrophages caused phagolysosomes to form and led to suppression of *M. tb* infection [35]. During *M. tb* infections there is increased IFN-γ to signal autophagy and translocate *M. tb* bacterial products to lysosomes for degradation [35]. With this knowledge, targeting the pathways involved in autophagy, like Vitamin D receptor signaling, different nuclear receptors, AMPK pathway and other small molecules can have potential benefit in fighting and protecting from this infection [36]. In summary, it has been shown that suppression of autophagy in *M. tb*-infected macrophages leads to increased bacterial load whereas activation of autophagy pathway leads to increased bacterial killing decreasing load [35].

Another instance in which intracellular maintenance can play a role in infection control is mitophagy. In mitophagy, mitochondria can become tagged and recognized by intracellular autophagosomes which eventually leads to delivery to lysosomes [37]. Mitochondria have developed numerous mechanisms to deal with the high rate of ROS that mitochondrial DNA is exposed to [37]. The repair mechanisms for mitochondrial DNA are mainly achieved by polymerase gamma (POLG) that has exonuclease and base excision repair functioning [37]. Other supportive repair mechanisms include antioxidants, protein folding and degradation [37]. When mitochondria have become damaged or inefficient, they can then undergo mitophagy [37]. Mitophagy is a form of selective autophagy that is seen in the regulation of proteins, mitochondria, other organelles, bacteria, and viruses [37]. The recognition of these mitochondria to become targeted for mitophagy is a ubiquitin dependent and independent manner [37]. In the ubiquitin-mediated interaction with the LC3 adapter, PTEN-induced putative kinase 1 (PINIK1) and Parkin (PARK2) first discovered as genetic factors in Parkinson’s disease where mitochondrial dysfunction was seen [37]. In the ubiquitin independent interaction with LC3 adapter proteins, choline dehydrogenase (CHDH) is one of the molecules responsible [37]. CHDH is found within the mitochondrial membranes and when those membranes become damaged or disrupted, it interacts with a molecule p62 forming a complex that induces mitophagy processes [37]. Among these pathways and other just as complex ones, this is one way the body can induce a physiological response to protect the Integrity of the cell and its metabolism [37]. Studies suggest that compromised mitochondrial function leads to increased susceptibility and severity of *M. Tb* disease [38]. Seeing as mitophagy’s function is to allow for mitochondrial turnover in the setting of dysfunctional or damaged mitochondria, it is logical then to hypothesize that increased mitophagy can improve overall mitochondrial function within a host [38]. As we age, mitochondrial stability and function decline, leading to numerous inefficiencies in various systems throughout body [39]. There is still much to uncover on the therapeutic potentials of modulating this pathway, however, there have been some promising results showing that in doing so can lead to decreased replication of *M. tb* [38].

One possible such pathway to be targeted is NIX-mediated mitophagy pathway. NIX knockdown abolished the induction of LC3 adaptor proteins upon infection with BCG [40]. Their experiment showed that NIX knock down had also altered metabolic reprogramming within immune cells [40]. It also showed there was significant upregulation of mitophagy receptor molecules leading to increased selective mitochondrial degradation [40]. NIX was seen to be upregulated in *M. tb* infection and BCG leading to increased cellular metabolism [40]. When NIX was knocked down it also negatively affected the production of TNF-α, IL-6, IL-1β [40]. These cytokines are secreted by macrophages as a necessary signal in effectively defending the body and increased bacterial clearance [40]. In summary, we can see that these crucial different pathways in mitophagy and can ultimately play a role in mycobacterial infection protection [40]. Figure 2 and Figure 3 below demonstrates autophagy and mitophagy pathways.

### 5.5. mTOR Signaling

Mammalian target of rapamycin (mTOR) is a major signaling pathway involved in growth, metabolism, and regulation of certain gene’s expression [41]. Normally, it’s dysfunction can lead to certain metabolic diseases such as obesity and type 2 diabetes as well as cancer or certain neurological diseases [41]. The pathway behind the mechanism is extremely complex and involves many proteins, factors, and genes. In essence mTORC1 is a complex that acts downstream on transcription factors [41]. These factors lead to an upregulation of glycolysis and glucose uptake. mTOR also acts in the autophagy/mitophagy pathways [42]. As previously mentioned, autophagy is allowing for the removal of damaged macromolecules and organelles [42].

It has been concluded that there is an inverse relationship between mTOR activation and autophagy induction [42]. When mTORC1 becomes activated, it inhibits numerous complexes within the cell, including autophagy [42]. Proteins in this pathway that become lead to less nucleation and transcription of the autophagosome pathway factors [42]. mTOR has also been shown to modulate certain mitochondrial functions. mTORC1 stimulates several mitochondrial regulators like TFAM, ribosomal proteins and components of electron transport chain complexes by upregulating nuclear components and increasing mRNA translation [43]. Inhibition of mTOR has been shown to limit the amount of mitochondrial biogenesis and cellular respiration [43]. In mice, when mTOR was inhibited, it amounted to decreased oxygen consumption, decreased motor activity, and decreased heat production [43]. mTORC2 also plays its own specific role in mitochondrial function [43]. If mTORC2 has decreased functioning, it leads to disruption of the mitochondrial membrane stabilization allowing an increase in membrane potential and calcium influx [43].

As previously mentioned, mTOR activation has an inverse effect on autophagy and mitophagy [44]. In ageing, as autophagy is already on a natural decline which leads to an increased amount of oxidative stressed, damaged organelles and damaged proteins [44]. Therefore, inhibition of mTOR in an ageing population could lead to increased amounts of autophagy, allowing for potential increased clearance of old and dysfunctional cells and mitochondria. Furthermore, as ageing progresses, stem cell and immune cell function declines [44]. Immune function is also generally declined as we age and is crucial because it’s our ability to be able to fight infection further along in our life [44]. mTORC1 inhibition is used as therapy as an immunosuppressive to limit T-cell activation following transplants. Inhibition of mTORC1 also augments the CD8^+^ response [44].

In *M. tb* infections, cells that are infected show an accumulation of ubiquitin inside the autophagosome allowing fusion between it and the lysosome inducing pathogen killing [45]. Part of the reason survival of *M. tb* infection occurs is from the result of disrupted fusion during this process [45]. mTOR’s activity with *M. tb* infection has been shown to be able to be modulated. Specifically, *M. tb* has been shown to increase mTOR activity on metabolism enhancing aerobic glycolysis which is thought to be a dominant component of mounting a sufficient immune response against *M. tb* infection [46,47]. So, for example, anyone with impaired glucose tolerance or metabolism would have a reduced capacity to mount necessary metabolic needs to fight the infection [47]. Additionally, due to the nature of granulomas in the defense of *M. tb* infection, there is an increasingly hypoxic environment within the walls of these granulomas. As a result, this puts stress on the immune cells within leading to an inhibition of mTOR and ultimately an upregulation of autophagy mechanisms and therefore increased destruction of *M. tb* organisms [46]. Figure 4 shows mTORC pathways and their effect on metabolism and cell growth [45].

## 6. Detecting *M. tb* Infection

The clinical identification of latent-type *M. tb* involves two commonly used assays which include the Interferon-Gamma Release Assay (IGRA) and the purified protein derivative (PPD) test administered through the Mantoux tuberculin skin test (TST) method, a technique developed by Charles Mantoux in 1912 [48]. In areas of low income, TST is more economically feasible. On the other hand, in middle- or high-income areas, either the TST or IGRA approaches of latent *M. tb* testing are recommended by the World Health Organization (WHO) [49].

The PPD test contains an extracted tuberculin protein from *M. tb* [50]. The Mantoux TST method involves an intradermal injection of 5 tuberculin units (5 TU) of standardized PPD (PPD-S) or 2 TU of PPD RT23 [48]. The former is used more in the United States (U.S.) while the latter is utilized widely within a global setting [50,51,52]. Within 24–72 h of inoculation, erythema and induration are observed at the injection site.

Nevertheless, a positive TST yields low species specificity. A major limitation of TST is that it can only identify a history of an immune response to *M. tb*. Infection, rather than, active infection [53]. Furthermore, a positive TST cannot distinguish between a prior *Mycobacterium bovis* (BCG) vaccine, *M. tb*., or another species of mycobacterium infection [53]. IGRA overcomes TST’s lack of specificity through its ability to quantify the release of interferon-gamma (IFN), which is secreted from sensitized T-cells to the mycobacterium antigens of *M. tb* [54]. Additionally, IGRA was shown to better predict progression into active *M. tb* than TST within susceptible populations in endemic countries [53]. IGRA utilizes the ELISpot test (enzyme linked immunospot assay) or the ELISA (enzyme linked immunosorbent assay) to quantify the T cells secreting IFN-gamma or concentration of IFN-gamma, respectively [54]. Following blood collection and processing, a positive test is indicated if there is a quantification of ≥ 0.35 IU ml-1 indicating a likely reaction with *M. tb* antigens [54]. The Mantoux TST heavily relies on an intact cellular immunity to antigens to promote a type 4 delayed hypersensitivity reaction occurring after injection within 48 to 72 h. This response is mediated by T lymphocytes through the cell-mediated immunity [55]. However, both IGRA and TST are reported to have reduced sensitivity in immunocompromised populations [48,53]. This may certainly hold true within aging populations without a robust immune response. Anergy is a plausible mechanism for the reduced immune response to tuberculin antigens observed within aging populations receiving TST [56]. 

## 7. Ageing and Tuberculosis

Tuberculosis (TB) is a potentially deadly infectious disease caused by *M. tb* that affects millions worldwide and is especially damaging to the developing world. Each year, there are an estimated 10 million individuals who become ill with *M. tb* and 1.5 million who die [57]. Tuberculosis is especially deadly to the elderly due to a decline in immune function and difficulties with diagnosis due to the potential for lessened response to diagnostics such as a tuberculin skin test resulting in false negative results—and absence of visible symptoms [4]. It should also be noted that in the elderly, the hindered immune response is not so much a consequence of individual T-cells having a hampered response to the bacterium, but a systematic issue with dendritic cells. The hampered T-cell response is theorized to be resultant of a shortcoming of antigen-presenting cells, which can be diminished in the elderly [4,58]. With an increasing incidence rate and the rise of multi-drug resistant strains, there is an ever-present impetus to develop a wide range of effective treatments against *M. tb* [59].

The normal Immune response to *M. tb* involves the innate immune response and the adaptive immune response. Firstly, the innate immune response involves several elements in the body that are already present to combat the infection. When the airway is exposed to the bacterium, its epithelial cells act as a physical barrier to the entrance of the bacterium further into the body and signal to the rest of the body to produce inflammatory cytokines, IFN-γ, and TNF-α. Macrophages, another component of the innate immune system, can effectively combat the bacteria in some cases but also allow it to grow in number. Upon exposure to the bacterium, the body will also stimulate neutrophils and monocytes as well as control the inflammatory response, though neutrophils may also stimulate tissue damage as part of their response.. Finally, the body will also mobilize dendritic, mast, and natural killer cells to modulate the body’s inflammation and provide an appropriate immune response. As part of the adaptive immune response, the body secretes cytokines and engages antigen-specific T-cells which promote further control of bacterial multiplication. The body will also form granuloma to isolate the bacterium from the rest of the body and protect the body from further harm [26]. Impaired T-cell functions in ageing result in decreased to fend off *M. tb* infection.

### 7.1. Ageing Effect on Mitochondrial Function

Ageing erodes away at the immune system in various ways including disruption of its structure and function which can result in susceptibility to infection, chronic inflammation, elevated cancer incidence, increased autoimmunity, and decreased protection from vaccination [60]. One theory that has been widely debated concludes that ageing results from accumulation of ROS which induce cellular damage and an organism’s ability to overcome these changes [6]. The cellular damage that occurs from these ROS is thought to damage cellular mitochondria and mitochondrial DNA (mtDNA). This affects transcription and replication of mtDNA leading to impaired mitochondrial function causing a greater buildup of ROS, resulting in a never-ending loop until organ failure and cell death.

The cellular structure and function of mitochondria are crucial In regulating apoptosis [6]. Oxidative damage to the mitochondrial membranes can result in cristae remodeling and release of cytochrome *c*. Cytochrome *c* can then activate caspases in the cytosol inducing apoptosis [61]. An age-related buildup of ROS may contribute to this apoptotic pathway. Furthermore, of various proteins found on the mitochondrial membrane, adenine nucleotide translocase, was recognized to have a greater age-related oxidative change [62]. Impaired mitochondrial function decreases the available adenosine triphosphate (ATP) needed for adequate cell duty. ATP is a cell’s energy “currency”, made up from a nitrogenous base, ribose sugar, and three bonded phosphate groups [63]. Hydrolysis of one of the three bonded phosphate groups provides energy for cellular processes including but not limited to signaling, synthesis, transport, and muscle contraction [63].

Cells can undergo death via two pathways: apoptosis or necrosis. These two pathways are complex and utilized by many different organisms. Intracellular ATP is used to determine which pathway a cell will endure [64]. A study using jurkat cells found that ATP deficient cells, with access to glucose and able to complete glycolysis produced sufficient ATP to complete apoptosis. Moreover, glucose depleted cells treated with oligomycin did not provide sufficient ATP needed for apoptosis and underwent necrosis instead [64]. Increased levels of ROS in cellular mitochondria causes mitochondrial dysfunction and decreases ATP synthesis. This process ultimately results in cell death, organ failure, and organism demise.

As discussed previously, our immune system is made up of both innate and adaptive immunity [65]. Both of which utilize different cells and their functions to rid an organism of a particular contamination. Each of these cells contain mitochondria to produce the necessary energy needed for effective function. Mitochondrial dysfunction by ROS as seen with age results in decreased ATP production, signaling affect, and immune system demise. This in turn increases susceptibility to infection, autoimmunity, lack of protection from vaccination, and increased incidence of cancer [60]. All of which are more common among the ageing population and is no different regarding *M. tb* infection.

### 7.2. Ageing within the T-Cell System

In a study, exposing the effects of ageing on human lymphocytes subsets, it was found that as an individual ages, there is a significant decline in the number of CD8^+^ T-cells [66]. As discussed previously programmed death-1 (PD-1) pathway appears to play a role in T-Cell exhaustion [12]. Additionally, evidence that the process of T-Cell exhaustion may share a similar pathway as T-cell ageing has also been suggested [67]. In a study designed to observe the epigenetic effects of ageing on CD8^+^ T-Cells, it was discovered that CD8^+^ T-Cells undergo an age-related loss of chromatin at promoter cites leading to a decrease in promoter binding of nuclear respiratory factor-1 (NRF1) [68]. NRF1 is a transcription factor known to hold a key role in regulating mitochondrial genes expression [68]. Loss of NRF1 could likely explain the loss of mitochondrial genes needed to complete oxidative phosphorylation an ATP generation [67]. This loss of ATP generation can ultimately lead to cell necrosis and cell death. The connection between T-cell exhaustion and T-cell ageing can be explained by PD-1 repression of NRF1 cofactor Peroxisome proliferator-activated receptor gamma coactivator 1-alpha (PGC1-a) resulting in decreased expression of NRF1, and mitochondrial respiratory genes seen in T-cell ageing [69].

Another pathway shown to be involved in T-cell ageing is immune senescence [14]. Senescent T-cells are characterized by shortening of telomeres, increased DNA damage from phenotypical changes, and cell cycle arrest ultimately leading to cell death [70]. Telomeres are defined as specific DNA nucleotides found at both ends of a chromosome that protect the genome from nucleolytic degradation and other damage [71]. With each round of normal cell division, the telomeres, shorten leading to chromosomal genome exposure, DNA damage, and permanent cell arrest; known as T-cell senescence [72]. DNA damage that arises regarding chromosomal exposure has been shown to alter key genes including genes coding for p53, p21, p16, Cdk2, Cdk6 and cyclin D3 [70]. The dysregulation between these genes leads to cell cycle arrest and the senescence seen with age. Senescent T-cells due to ageing result in a decline in the number of mature T-cells leading to impaired response to infection [73]. The impaired ability of the innate and adaptive immune system to recognize *M. tb* antigens has been theorized to play a role in the susceptibility to this bacterial infection [74]. Data has been found that indicates peripheral blood mononuclear cells from the ageing population are found to generate reduced amounts of interferon gamma (IFN-γ) from CD4 T-Cells, ultimately leading to impaired immune response, specifically to *M. tb* infection [74].

On the contrary, evidence of no difference between adult and elderly T-cell response to *M. tb* has also been observed. According to a study on T-cell immunity in the elderly, it was perceived that neither adults nor the elderly showed a statistical difference in the frequency of production of interferon-gamma (IFN-γ) regarding latent and active *M. tb* infection [17]. Given this information, support for an entirely different theory has gained traction in that the increased susceptibility to *M. tb* infection seen in the elderly could in fact be due to altered monocyte phenotypes and their ability to phagocytose infectious pathogens [17]. While although the individual T-cell response to infection may not be affected by old age, the overall lack of T-cells numbers in the elder population results in a weaker immune system response collectively.

### 7.3. How Ageing Affects mTOR Signaling, Autophagy & Mitophagy

Ageing has been associated with certain “hallmarks” including but not limited to; genomic instability, telomere degradation, epigenetic alteration, loss of proteostasis, deregulated nutrient-sensing, mitochondrial dysfunction, cellular senescence, stem cell exhaustion, and altered intercellular communication [75]. mTOR, as stated previously, is a major signaling pathway involved in growth, metabolism, and regulation of certain gene’s expression [41] and is known to influence a number of these ageing “hallmarks” [76]. Furthermore, ageing in general has been shown to over activate the mTORC1 pathway [77]. The mTORC1 pathway is responsible for the inhibition of autophagy [77]. Autophagy has been shown to slow ageing by clearing damaged cells within an organism [77]. Therefore, by increasing the activation of mTORC1 pathway seen in ageing populations, a decline in autophagy is observed [77]. Moreover, a decrease in autophagy will result in a buildup of misfolded proteins and damaged organelles within an organism that have been shown to accompany age-related diseases such as Type 2 Diabetes Mellitus, Parkinson disease, and even Alzheimer disease [77].

In mitochondria, mitophagy is the process in which a buildup of misfolded proteins and damaged cells are eliminated [77]. Furthermore, ageing affects this process via overactivation of the mTORC1 pathway as well. The buildup of misfolded proteins within the mitochondria, due to lack of mitophagy, leads to mitochondrial dysfunction. Mitochondrial dysfunction is defined as abnormal morphology, lack of ATP synthesis, widespread production of ROS ultimately leading to mutations in mtDNA [78]. As discussed previously, an increase in ROS leads to an increase in mutations in mtDNA and an increase in apoptotic pathways [61] resulting in lack of ATP synthesis [6]. ATP is needed to provide energy for cellular processes including but not limited to signaling, synthesis, transport, and muscle contraction [63]. Reduced ATP leads to cell death, organ failure, and organism demise.

A study conducted in 2015 found that mTORC1 phosphorylation and therefore activation was significantly higher in older adults than in younger adults irrespective to gender [79]. Provided this information, it should be hypothesized that mTOR induced lack of autophagy and mitophagy seen in the ageing population suppresses an individual’s ability to fight infections such as *M. tb*. Figure 5 illustrates the effects of aging on mitochondrial ATP production, protein stability and organelle damage from impaired autophagy.

### 7.4. Ageing Effect on GSH System

GSH, as defined earlier, plays a crucial role in host antioxidant defense, nutrient metabolism, and regulation of many cellular processes [10]. Research has shown that the GSH redox system is, in fact, affected by age resulting in dysfunction of the above processes [80]. Studies have also shown an association between age and decreased levels of GSH and GSSG which may lead to impaired immunological response, increased ROS, and greater susceptibility to infectious diseases such as *M. tb* [80]. It has been demonstrated that levels of GSH were significantly diminished from young (1–11 years) to old (41–69 years) individuals and peaked around middle age (25–45 years) individuals [80]. Data from a study designed to measure the levels of GSH in aged rat brains indicates the GSH decline was a result in both “the reduction of GCS-mediated de novo GSH synthesis and the increase of consumption in aged brain” [81]. The age-related decline in GSH synthesis has also been theorized to include “increased rate of oxidation in senescence, decreased GSH synthesis due to cysteine deficiency, and/or diminished activity of glutamylcysteine synthetase and increased GSH consumption in the removal of peroxides and xenobiotics” [80].

Among the numerous processes affected by GSH, one critical role is its inhibition of intracellular *M. tb* growth [82]. It has been hypothesized that GSH is able to achieve bacterial growth inhibition due to its ability to bind the same D-alanyl-D-alanine-carboxypeptidase (DacC)-gene site as ampicillin thus resulting in decreased DacC enzyme transcription [83]. DacC enzyme is, considered to be, one of the enzymes responsible in peptidoglycan polymerization and therefore a decrease in its transcription will subsequently result in decreased peptidoglycan and cell wall synthesis [83]. Given the antibacterial mechanism with which GSH is hypothesized to possess, it should be assumed that a decrease in GSH results in favorable conditions for *M. tb* growth. Further testing should be directed towards studying whether an increase in exogenous GSH, in an age-related population, can help prevent *M. tb* infection.

### 7.5. Genes/Transcription Factors That Play a Role in Ageing

Outside of the many mechanisms laid out prior, there are many specific genes and transcription factors that can play a role in the protection of the cell. One such gene is nuclear factor erythroid2-related factor 2- antioxidant response element (NRF2-ARE), which is activated in the present of highly reactive electrophiles [84]. In a physiologic state, cells normally sequester NRF2 in the cytosol via Keap1 proteins, leading to polyubiquitination of the complex and eventual proteasomal NRF2 degradation [84]. However, for NRF2 to be freed from its Keap1 complex, several stressors can be present [84]. Several main pathways of activation have been described, first of which is the presence of ROS [84]. Cysteine residues found on the Keap1 proteins can be modified in the presence of ROS or other reactive electrophiles leading to a release of Zinc turning Keap1 into a confirmation no longer able to bind NRF2 [84].

The next proposed activation pathway is due to cell regulator molecules like p62 and p21 bind Keap1 with higher affinity than that of NRF2, therefore allowing NRF2 to be released and eventually translocate to the nucleus [84]. Additionally, there are many different intracellular protein kinases that can modify NRF2 itself, changing its ability to be sequestered by Keap1 [84]. Lastly, when AKT is upregulated, it leads to inhibition of GSK3, leading to no phosphorylation of NRF2 keeping it in its active state [84]. Ultimately, this group of activation pathways leads to free NRF2, leaving it to translocate to the nucleus and bind ARE elements in DNA forming a dimer [84]. This dimer allows for transcription of genes of numerous antioxidant proteins like Cystine/glutamate exchanger, γGCL, GPX, GR, redox ins and many more [84]. Importantly it is linked to GSH synthesis and stability because of its role in creating GSH synthase and GSH reductase [84]. Hence, in an ageing population who already has a more difficult time in managing oxidative stress and infection mentioned previously, the genes responsible in creating the systems to protect cells in the first place also seem to be found in decreased levels [85]. As illustrated in numerous studies, *M. tb* is seen to have increased levels of oxidative stress and lower levels of antioxidants, like GSH [86]. As discussed previously, NRF2 levels increase in times of rising levels of ROS, so intuitively we can see increased NRF2 in *M. tb* infections also [86]. However, the increased levels of NRF2 were found to be within the cytoplasm, which leads us to believe that there was less nuclear activation despite increasing cytoplasmic levels [86]. In a mouse model experiment, the role of NRF2 was able to be confirmed by way of the results showing that that in NRF2 deficient mice infected with *M. tb* had significant drops in granuloma formation with significant higher levels of macrophage activating cytokine like IL-2 and IL-13 [86]. Overall NRF2 deems to be protective in *M. tb* infection, however, becomes more complicated when looking at early versus late stages of infection [86]. The effects of ageing previously mentioned above is summarized in Table 2 below and the pathways are summarized in Figure 6.

## 8. Ageing Effects on Susceptibility to *M. tb* Infection and Possible Preventative Strategies

Through the numerous mechanisms discussed in the defense against *M. tb* infection, we can see that as the ageing process occurs many of those mechanisms can become disrupted. T-cells need finely tuned and regulated homeostasis while maintaining a quiescent state, but with ageing the effect is seen more with intrinsic activation of normal pathways leading to defective signaling withing the cell [87]. mTOR, autophagy, mitophagy are all inner linked as previously discussed, and once again, as our cells age and the signaling pathways start to become less efficient, we have a complicated interplay between mTOR, auto phagocytic processes and their role in *M. tb* infection defense [44,76]. GSH, a major antioxidant pathway has been deemed significant in the defense in *M. tb* infection, and because of ageing, the production of GSH is lowered allowing for ROS to damage cells and during infection [32,80,82].

In summary, as we age many of our body’s innate systems start to lose their peak functionality and leave gaps for adverse events to take place, like the increased susceptibility to *M. tb* infection. With this knowledge, we can hypothesize and develop strategies to modulate and mitigate some of the damaging effects of ageing on the immune system. One such strategy is the supplementation of GSH [88]. In one such human trial, where HIV-positive individuals were given GSH supplementation, [88] the study concluded that with the supplementation, GSH levels were able to be increased and restored, leading to increased production of IL-1β, IL-12, and IFN-γ, decreased the levels of IL-10, TGF-β, and IL-6 [88]. Therefore, the increase in these cytokine combinations infers a favorable immunological environment an improved function at controlling a potential *M. tb* infection [88].

Another strategy that has been utilized for over 80 years is the administration of the Bacille Calmette–Guerin (BCG) vaccine [89]. During a cross-sectional study of *M. tb* in the elderly, has revealed that obtaining the BCG vaccine at birth indeed has protective factors and constitutes lower odds of becoming infected and dying from *M. tb* infection [4]. Moreover, according to the Center of Disease Control (CDC) use of the BCG vaccine as a preventative strategy is recommended only in populations who meet a specific criterion [90]. As time has gone on, the effectiveness and implementation of the BCG vaccine have gone under revaluation in the hopes of improving outcomes of those who have received the vaccine [89]. Various changes to the vaccine composition, booster scheduling, and even the route of administration are just some of the avenues that have been explored in the process of reevaluating how the BCG vaccine can be improved. Nonetheless, more research is needed to fully understand the benefit weighed against any potential risks of the BCG vaccine. However, given our current information on BCG vaccination, it is our recommendation to continue utilizing the BCG vaccination as a preventative measure in the fight against *M. tb* infection [89].

## 9. Conclusions

In summary, ageing has numerous effects on various biochemical, physiological, and immunological processes. As we have briefly outlined in this paper, those effects on the various processes can be detrimental to many aspects of life, but our body’s ability to protect and properly defend against *M. tb* infection. Some of the processes can have possible therapeutic targets and potentially could lead to a more robust defense against *M. tb* infections. While there is still much more to learn and understand, given the information we know, it would be reasonable to consider administering the BCG vaccine alongside GSH supplementation to act as potential therapy in the ageing population and helping to defend and mitigate the effects of *M. tb* infection. More research would have to be undergone, especially with the changing landscape of the BCG vaccine, in novel immune therapies, and *M. tb* infections.

## Figures and Tables

**Figure 1 vaccines-10-01892-f001:**
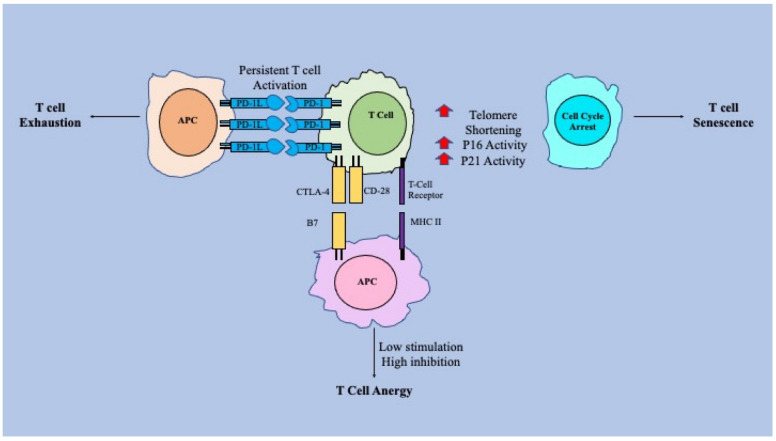
Ageing-associated mechanisms of T-cell dysfunction.

**Figure 2 vaccines-10-01892-f002:**
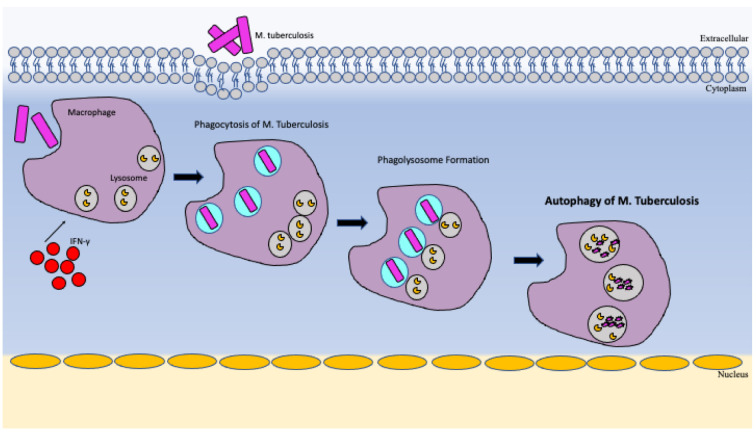
Autophagy of *M. tb* to eradicate and prevent growth.

**Figure 3 vaccines-10-01892-f003:**
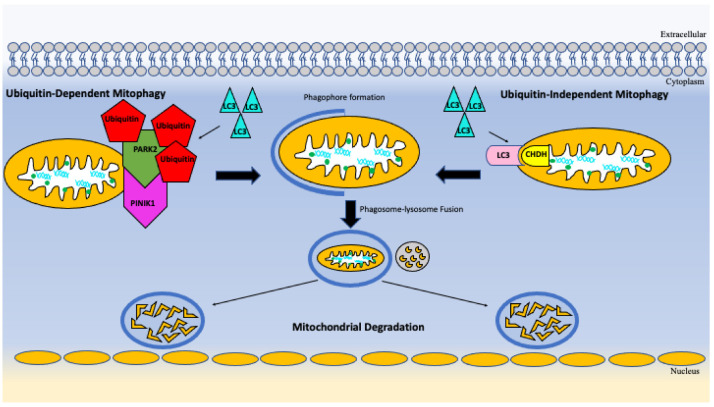
Mitophagy pathways demonstrating mitochondrial turnover.

**Figure 4 vaccines-10-01892-f004:**
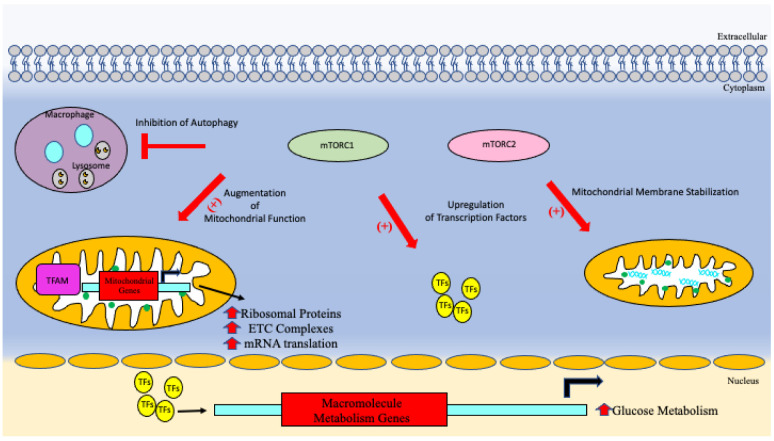
mTORC1 and mTORC2 pathways on autophagy, mitochondria, cell growth, genetics, and mitochondrial membrane stabilization.

**Figure 5 vaccines-10-01892-f005:**
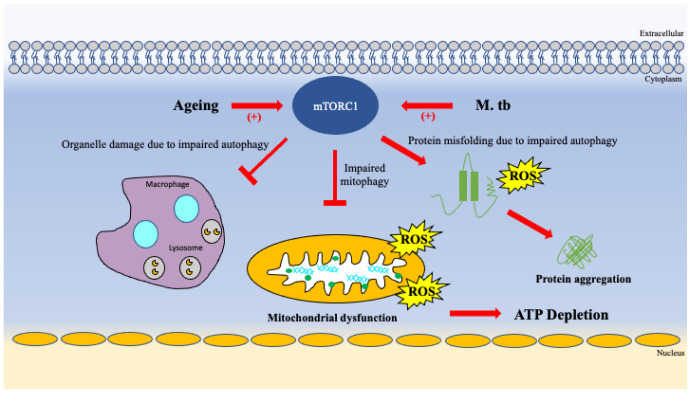
Ageing impairs ATP production, exacerbates buildup of protein aggregates and organelle damage.

**Figure 6 vaccines-10-01892-f006:**
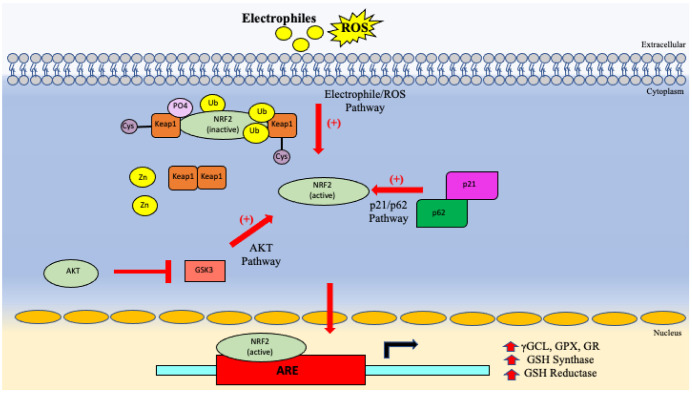
Pathways that affect the major genes which promote or affected in aging.

**Table 1 vaccines-10-01892-t001:** Key mediators of the immune response to *M. tb*.

Mediators	Mechanism of Immune Response
Macrophages	Macrophages are the first responders to infection. Recognition of *M. Tb* occurs through TLR and PAMP interaction. This leads to activation of NF-κB to upregulate mediators that promote inflammation and the immune response [21,22].
T-Cells	Both CD4^+^ and CD8^+^ play a role in preventing the spread of *M. tb* throughout the body. CD4^+^ cells promote antibody production and increase the recruitment of CD8^+^ cells. Those found to have lower counts of CD4^+^ are associated with an increased risk for infection [26].
Cytokines	Important cytokine mediators include IFN-γ which accentuates macrophage recruitment, while TNF-α, IL-1α and Il-1β promote the host inflammatory response. [22,23,24].
Reactive Oxygen Species	The host defense with reactive oxygen species begins with the production of superoxide radical O^2-^ via the NADPH oxidase pathway which promotes control and elimination of *M. Tb* [24].
Autophagy	Mediates intracellular degradation to decrease bacterial load. Upon infection, there is an upregulation of ubiquitin-dependent autophagy of *M. tb* [21,24].

**Table 2 vaccines-10-01892-t002:** Summary of ageing factors that increase susceptibility to *M. tb*.

Factors of Immunity	Mechanism of Immune Dysfunction
ATP Production	ROS accumulation leads to mitochondria dysfunction, depleting ATP generation within cells that are imperative to the immune response [6,47,49].
T-cell responses to infection	CD8 T-cell exhaustion and senescence exacerbated in ageing leads to a suboptimal response to infection [11,13,53,60].
Cell clearance and turnover	Overactivation of mTORC1 signaling impairs autophagy and mitophagy resulting toxic build up cellular waste products, misfolded proteins, and damage organelles [63,64].
Defense against oxidative stress	Diminished levels of GSH through reduced GSH synthesis attenuates the immune response in aging populations, increasing susceptibility to infection [47,67].
Genes promoting cell protection	Decreased gene expression of NRF2 reduces GSH synthase and reductase levels which promotes immunosuppression and the inability to neutralize oxidative stress [67,71].

## Data Availability

Not applicable.

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
