# Peer review of "Mycobacterium tuberculosis: Implications of Ageing on Infection and Maintaining Protection in the Elderly"

_vaccines, 2022, doi:10.3390/vaccines10111892_

Round 1

Reviewer 1 Report (Previous Reviewer 2)

The manuscript has now been revised in a satisfactory way.

Author Response

Dear Reviewer#1,

We appreciate your positive comments and constructive feedback.

We look forward to the acceptance of this manuscript.

Comment: The manuscript has now been revised in a satisfactory way.

Response: Thank you for this feedback, we are pleased to hear our edits and comments were satisfactory.

Reviewer 2 Report (New Reviewer)

Minor comments

The review Bonavida group discusses the effect of ageing in tuberculosis. The article is intriguing and very well written. I have a few minor concerns though.

1> Authors did not mention anything about granuloma formation in the lungs during tuberculosis.

It should be mentioned and discussed with updated references though it is not the innate immune compartment.

2>”Line 22, 31”- “risk of dying”- please change this vulnerable phrase.

3> Line “133”-“T-Cell System definition”- This term does not make any sense. It is requested to be modified. In this section, authors discussed about CD8+ T cell exhaustion and anergy. Authors should also discuss what could be the effect of those during tuberculosis with proper citations. Are

those well studied in the tuberculosis research realm?

4> CD4+ and CD8+ T cells, the “+” sign should be “superscript” all over the manuscript.

5> “Line 231”- “They way these cells mediate this”- It does not make any sense.

6> “Line 433”-“A falsely negative skin test”- Authors should refrain from such negative approaches. In addition, currently IGRA (Interferon-Gamma Release Assay) is the gold standard for tuberculosis detection. Authors should mention about IGRA and might discuss the shortcoming of Mantoux tuberculin skin test (TST) with updated references.

Author Response

Dear Reviewer#2,

We appreciate your positive comments and constructive feedback.

We look forward to the acceptance of this manuscript. 

Comment 1: Authors did not mention anything about granuloma formation in the lungs during tuberculosis. It should be mentioned and discussed with updated references though it is not the innate immune compartment. 

Response 1: Thank you for your feedback on this. We completely agree that granulomas are a huge component of the immune response in the context of M. tb. So we added an entire new section (Section 4: M.tb and Granuloma formation). This section discusses the basics and overview on granuloma formation and its roles and is highlighted within the manuscript. We have also provided the new addition text below in red.

Granuloma formation is a process that provides protection to the host by preventing replication and spread of M. tb infection. However, in the last decade a lot of research has been done showing that this process and its role is not as clear cut and has many nuances to it. The reason for this is the conflicting presentation of the type of role granuloma formation plays in infection. On one side, the process seems to provide a vessel for bacterial expansion and replication while on the other side this process seems to allow for sequestration and protection of bacterial products (Pagan).

            There is some research that shows that pathogenic forms of M. tb can hijack and manipulate host immune responses to take leukocyte and cell death processes to allow for bacterial replication and expansion. ESX-1, a bacterial secretion system is a gene that has been shown to play a role in the virulence as shown in vitro of human cells infected with M. tb. It was discovered that the region of difference 1 (RD1) locus contains the sequences for ESX-1 genes. It was shown that changes or mutations in RD1 showed a defect primarily in granuloma formation and showed decreased bacterial numbers suggesting that granuloma’s role points more towards expansion of bacteria rather than protection when compared to wild-type M. tb strains (Pagan).  

            On the other hand, other researchers have suggested that while M. tb may be able to induce and use granuloma formation for its own expansion, others say that the granuloma formation can serve protective functions as well. While these mechanisms are not fully understood and hard to decipher, there are some points made that would indicate this can be true. For example, granulomas contain T-cells within them and help transfer protective immunity to within the walls of the granuloma. Hence, inherently there are protective cells found within the granuloma itself that we know play a critical role in inhibiting M. tb infection. Furthermore, in those populations, like the elderly, with decreased T-cell immune responses, macrophages cannot be properly activated by T-cells. Another example is one shown in a study of HIV infected individuals who had concomitant M. tb infections. HIV individuals have decreased granulomatous responses and in turn decreased control of M. tb infection (Ehlers).

            Overall, it is possible that mycobacteria might benefit from the granuloma formation process and whether there is any protective function has truly been left unproven. However, there is the data supporting that the formation of them do indeed provide a bacterially favorable environment to allow for expansion. Hence, when looking at possible treatment options, especially in elderly populations, slowing of granuloma formation should be considered (Pagan). At the same time, the inherent possible protective functions of granuloma formation and action also should be considered and studied further (Ehlers).

Comment 2: ”Line 22, 31”- “risk of dying”- please change this vulnerable phrase.

Response 2: Thank you for this feedback, we agreed this phrasing can mislead and be misinterpreted outside of the context of what we are trying to accomplish. The phrasing was deleted and changed. 

Comment 3: Line “133”-“T-Cell System definition”- This term does not make any sense. It is requested to be modified. In this section, authors discussed about CD8+ T cell exhaustion and anergy. Authors should also discuss what could be the effect of those during tuberculosis with proper citations. Are those well studied in the tuberculosis research realm?

Response 3: Thank you for this feedback, the title of the section was changed to “Immune System”: T-Cells to move away from the previous title of T-Cell System Definition. Furthermore, we added a small section to the end of this same paragraph that discusses the various processes and their effect during M. tb infections. Please refer to new highlighted paragraph within the manuscript and also below for convenience. 

Not only are each of these three processes; CD8+ T cell exhaustion, anergy, and senescence, crucially affected by ageing, but also by chronic infections such as M. tb. It has been found that cells in individuals who have been infected with M. tb have been discovered to have aged prematurely, leading to senescent cells, directly correlating with an M. tb survivor’s high risk of mortality (Bobak C, Natarajan H). Additionally, anergy in patients with active M. tb infections has been thought to be a response to monocytes actively producing cytokines that effectively suppress lymphocytes and promote fibrosis such as transforming growth factor ? (TGF-?) (Wahl S.). Interestingly, a study has shown that CD8+ T cells, with a lack of coexpression of receptors, in individuals with M. tb has not shown effects of CD8+ T cell exhaustion (Rozot V) and therefore begs the question if ageing is the only factor affecting CD8+ T cell exhaustion? More research is needed to fully understand this connection.

Comment 4: CD4+ and CD8+ T cells, the “+” sign should be “superscript” all over the manuscript.

Response 4: Thank you for this feedback. Necessary formatting changes were and they should all be superscript now. 

Comment 5: “Line 231”- “They way these cells mediate this”- It does not make any sense.

Response 5: Thank you for this feedback, we have rectified the wording and a new sentence is provided below. 

They way CD4+ mount this type of defense and the antigens they bind to are not well understood

Comment 6:  “Line 433”-“A falsely negative skin test”- Authors should refrain from such negative approaches. In addition, currently IGRA (Interferon-Gamma Release Assay) is the gold standard for tuberculosis detection. Authors should mention about IGRA and might discuss the shortcoming of Mantoux tuberculin skin test (TST) with updated references.

Response 6:  Thank you for this feedback. We appreciated the nature of the verbage used in describing the negative skin test. The purpose of this sentence was to highlight the problem with using the skin tests in the elderly due to a diminished Type IV hypersensitivity reaction to the test itself because of the elderly poor immune response to the antigens. A section has been outlining the IGRA vs TST and is now labeled “Detecting M. tb infection”

Detecting M. tb infection

The clinical identification of latent-type M. tb involves two commonly used assays which include the Interferon-Gamma Release Assay (IGRA) and the purified protein derivative (PPD) test administered through the Mantoux tuberculin skin test (TST) method, a technique developed by Charles Mantoux in 1912 [48]. In areas of low income, TST is more economically feasible. On the other hand, in middle- or high-income areas, either the TST or IGRA approaches of latent M. tb testing are recommended by the World Health Organization (WHO) [49].

The PPD test contains an extracted tuberculin protein from M. tb [50]. The Mantoux TST method involves an intradermal injection of 5 tuberculin units (5 TU) of standardized PPD (PPD-S) or 2 TU of PPD RT23 [48]. The former is used more in the United States (U.S.) while the latter is utilized widely within a global setting [50]–[52].

 Within 24-72 hours of inoculation, erythema and induration are observed at the injection site.

Nevertheless, a positive TST yields low species specificity. A major limitation of TST is that it can only identify a history of an immune response to M. tb. Infection, rather than, active infection [53]. Furthermore, a positive TST cannot distinguish between a prior Mycobacterium bovis (BCG) vaccine, M. tb., or another species of mycobacterium infection [53]. IGRA overcomes TST’s lack of specificity through its ability to quantify the release of interferon-gamma (IFN), which is secreted from sensitized T-cells to the mycobacterium antigens of M. tb [54]. Additionally, IGRA was shown to better predict progression into active M. tb than TST within susceptible populations in endemic countries [53]. IGRA utilizes the ELISpot test (enzyme linked immunospot assay) or the ELISA (enzyme linked immunosorbent assay) to quantify the T cells secreting IFN-gamma or concentration of IFN-gamma, respectively [54]. Following blood collection and processing, a positive test is indicated if there is a quantification of >/= 0.35 IU ml-1 indicating a likely reaction with M. tb antigens [54]. The Mantoux TST heavily relies on an intact cellular immunity to antigens to promote a type 4 delayed hypersensitivity reaction occurring after injection within 48 to 72 hours. This response is mediated by T lymphocytes through the cell-mediated immunity [55]. However, both IGRA and TST are reported to have reduced sensitivity in immunocompromised populations [48], [53]. This may certainly hold true within aging populations without a robust immune response. Anergy is a plausible mechanism for the reduced immune response to tuberculin antigens observed within aging populations receiving TST [56].

This manuscript is a resubmission of an earlier submission. The following is a list of the peer review reports and author responses from that submission.

Round 1

Reviewer 1 Report

The authors select a very important and contemporary topic that is not well understood, so this is important for the literature.  However, the title of the review is largely disconnected or lost in the actual content.  Specifics are provided below.

Major comments:

Overall:

-        The text is largely devoted to an overview of selected mechanisms that influence the immune response to M. tuberculosis, with descriptions mostly on the mechanisms, and then a small component on how they may affect Mtb containment. Only starting in section 8 is old age brought up. But there is not an in-depth dive into what is known on aging and TB combined, which is what the title implies.  Restructuring and shifting the emphasis of the entire content is suggested.  If details on mechanisms are going to be kept, then many more figures, or more detailed figures with the described pathways are required. Otherwise it is very difficult to follow.

-        Given that TB is a pulmonary infection, studies relevant to altered immunity in this compartment in the context of TB should be brought up, even if it is just briefly. 

-        The writing has a number of grammatical mistakes (some misspelling also) that make the flow of the review difficult to follow. This, combined with in-depth description of pathways and molecules that are not accompanied by pictures, further challenges its comprehension.

-        Inflammaging. Role in TB risk. Not mentioned in abstract nor in the review, even though this is a hallmark of compromised immunity in aging.

-        Below I provide some specifics to the authors on these points, although the points described above relate to all the sections and not necessarily pointed out for each section.

Abstract.

-        Suggest clarification: The biggest risk of TB in the elderly is not so much the development of TB, but rather, their high risk of dying from it.  Inclusion of this latter aspect is important.

Introduction.

-        Please consider citing recent studies addressing the higher risk of the elderly to TB and death from TB.  Scordo et al, PlosOne 2021;

-        Please consider citing recent studies that show that the elderly do not have an evident defect in cell-mediated immunity to M.tb antigens.  Scordo, Aguillón-Durán et al, 2021;  Ault et al, 2018.  Important to clarify that the “T-cell anergy” in elderly patients based on a negative tuberculin skin test, is not a reflection of systemic T-cell anergy, but rather, a defect in the elderly in APC by their skin dendritic cells. The field has been misled for years based on this misconception.

-        TB statistics. There are about 1.4 million deaths per year, or where did the 2.9 million come from? There is no reference cited for this statistic and ref 15 is only for MDR-TB which is irrelevant.

-        The innate immune response to Mtb comprises additional components beyond neutrophils and monocytes. In fact, it is thought that in the lungs, the initial response is carried out by alveolar macrophages. There are also soluble components of the innate immune response, including anti-microbial peptides, among others. This section lacks depth and breadth.

Aging and tuberculosis

-        This section only has one sentence that briefly touches on this topic.  Based on the title, this should be the core of the paper.

T-Cell system definitions.

-        Line 87-88. What is meant by “depletion”.  This sounds like an extreme event which is not the case for immune cell subsets.

-        Remove “A few of these processes can be easily confused…”

-        The descriptions for T-cell exhaustion, anergy and senescence are done in the context of infectious diseases or cancer. A sentence or two on how they occur or their features with aging would be relevant to this review.

4. Normal Immune Response of Mycobacterium tuberculosis infections

-        What does “normal” mean?

-        This section is difficult to follow. A number of immune response effectors are listed and described, but it is difficult to gain an integrated picture. A suggestion is to reference a detailed review on the topic, and provide here a summary of key aspects of the response that contribute to containment, others that allow for Mtb to remain viable and latent, and finally the host factors that contribute to Mtb replication and disease. Indicate what is not known.

Autophagy, mitophagy:

-        Long paragraphs with lots of detail, but overall little relevance to Mtb in the elderly. Suggest that the focus is more towards the role of autophagy and mitophagy in Mtb containment, and then how this may be altered in old age.

mTOR signaling:

-        Section with lots of details on pathways. Very difficult to follow, including its relationship to aging.

8. Ageing Effects on Immune System Pathway

- I thought this section would describe the known defects in immune function in the elderly.  Instead, it focuses on mitochondrial dysfunction, how it leads to lower ATP, and how this broadly affects immune function and other processes. So the section title does not coincide with its content.

9. Ageing Within the T-Cell System

- 1st paragraph is difficult to understand

- Nice description of mechanism by which T-cell aging may occur.  However, with respect to TB, the summary of results is only one-sided, and data interpretation is more complex.  Only the defects in CD4 T-cells are summarized, but data on the early resistance conferred by CD8 T cells is not described. All these studies are in mice. Furthermore, there is no summary on other studies where no defects were identified in T-cell mediated responses of humans in response to mycobacterial antigens.  See Ault, Dwivdi et al, 2018; Scordo, Aguillón-Durán et al, 2021.  Consolidation of the different findings is important to provide a broader understanding.

10. How Ageing Affects mTOR Signaling, Autophagy & Mitophagy

- line 440. Do you mean “mitochondria”, and not “cells”?

- The 2nd to last sentence infers that mTOR-induced autophagy should increase risk of Mtb in the elderly. Suggest the authors provide examples of what is known in younger hosts with respect to this hypothesis. This would strengthen the case.  It would be of more value if the first paragraphs of this section be consolidated, and this last component should be expanded. 

- Figure 3. How may this affect TB susceptibility?  Also, this figure does not show autophagy, which is the topic of this section.

11. Ageing Effect on GSH System

- Nice section

12. Genes/Transcription Factors that play a role in ageing

- Needs a figure

13. Ageing effects on susceptibility to M. tb infection and possible preventative strate-520 gies:

- The recommendation for giving BCG vaccination is unclear. This topic is only brought up here and with brief foundation for its use. Of note, in Scordo et al, PlosOne 2021 report that no BCG vaccination at birth is associated with higher odds of TB.  However, these findings require further study before being considered as a basis for a recommendation for BCG in old age.  Data on BCG use and protection against COVID-19 has not supported its protective role for this viral infection.

Table 1.  

Overall unclear how this table fits with title of the study.  Where is the connection between aging and TB?

What does “ATP-dependent immune function" mean?  All immune function needs ATP.

Figure 1.  T-cell cytoplasm seems disproportionate (too big)

Author Response

Thank you for all of your feedback, comments, kudos and suggestions. The manuscript was extensively revised to incorporate all the recommendations and suggestions of the reviewers. The edits made are denoted as track changes. We hope to have improved this paper to your satisfaction and hope we addressed all the comments and concerns in a timely and knowledgeable manner. We look forward to the publishing this important work. Awaiting to hear back from you regarding the numerous changes that were made.

The authors select a very important and contemporary topic that is not well understood, so this is important for the literature.  However, the title of the review is largely disconnected or lost in the actual content.  Specifics are provided below.

Major comments:

Overall:

-        The text is largely devoted to an overview of selected mechanisms that influence the immune response to M. tuberculosis, with descriptions mostly on the mechanisms, and then a small component on how they may affect Mtb containment. Only starting in section 8 is old age brought up. But there is not an in-depth dive into what is known on aging and TB combined, which is what the title implies.  Restructuring and shifting the emphasis of the entire content is suggested.  If details on mechanisms are going to be kept, then many more figures, or more detailed figures with the described pathways are required. Otherwise it is very difficult to follow.

-        Given that TB is a pulmonary infection, studies relevant to altered immunity in this compartment in the context of TB should be brought up, even if it is just briefly.

- Discussed about alerted pulmonary immunity within the inflammaging paragraph.

-        The writing has a number of grammatical mistakes (some misspelling also) that make the flow of the review difficult to follow. This, combined with in-depth description of pathways and molecules that are not accompanied by pictures, further challenges its comprehension

           -We made additional table and figures. The revised manuscript now has six figures and two tables. Checked and corrected grammar and spelling mistakes.

-        Inflammaging. Role in TB risk. Not mentioned in abstract nor in the review, even though this is a hallmark of compromised immunity in aging.

            - Added a very detailed section on what inflammaging is, it’s effect on the immune system and how it translates over to M. tb infection in the elderly. Also added a brief part within the abstract.

-        Below I provide some specifics to the authors on these points, although the points described above relate to all the sections and not necessarily pointed out for each section.

Abstract.

-        Suggest clarification: The biggest risk of TB in the elderly is not so much the development of TB, but rather, their high risk of dying from it.  Inclusion of this latter aspect is important.

- We went ahead and added to the abstract that not only does ageing cause increased susceptibility to TB but also as a result, a higher risk of death once an older individual is infected given the diminished “collective” immune response.

Introduction.

-        Please consider citing recent studies addressing the higher risk of the elderly to TB and death from TB.  Scordo et al, PlosOne 2021;

-     The reference was added to the introduction and the section on ageing and TB

-        Please consider citing recent studies that show that the elderly do not have an evident defect in cell-mediated immunity to M.tb antigens.  Scordo, Aguillón-Durán et al, 2021;  Ault et al, 2018.  Important to clarify that the “T-cell anergy” in elderly patients based on a negative tuberculin skin test, is not a reflection of systemic T-cell anergy, but rather, a defect in the elderly in APC by their skin dendritic cells. The field has been misled for years based on this misconception.

-     This note was added to the section on ageing and TB

-        TB statistics. There are about 1.4 million deaths per year, or where did the 2.9 million come from? There is no reference cited for this statistic and ref 15 is only for MDR-TB which is irrelevant.

-     The reference was corrected, and the correct data was inserted.

-        The innate immune response to Mtb comprises additional components beyond neutrophils and monocytes. In fact, it is thought that in the lungs, the initial response is carried out by alveolar macrophages. There are also soluble components of the innate immune response, including antimicrobial peptides, among others. This section lacks depth and breadth.

-     The section on the immune response was expanded to include more aspects of both the innate and adaptive immune system.

Aging and tuberculosis

-        This section only has one sentence that briefly touches on this topic.  Based on the title, this should be the core of the paper.

- We realized where the disconnect may have stemmed from, after some formatting fixes within the manuscript, there should be clarity in that all the subsections following are influenced by age and how they affect the elderly on Tb infection. This paragraph serves as an overview and introductory paragraph with the following subsections having more detailed explanations. The paragraph was also moved farther down in the manuscript for better flow and readability of the article.

T-Cell system definitions.

-        Line 87-88. What is meant by “depletion”.  This sounds like an extreme event which is not the case for immune cell subsets.

            - Changed “depletion” to “diminution” to sound less extreme, but still describe processes in which T Cell function is diminished

-        Remove “A few of these processes can be easily confused…”

            - Changed this phrasing

-        The descriptions for T-cell exhaustion, energy and senescence are done in the context of infectious diseases or cancer. A sentence or two on how they occur or their features with aging would be relevant to this review.

            -Added a couple sentences addressing these pathways and how they affect or are affected in the presence of ageing.

  1. Normal Immune Response of Mycobacterium tuberculosis infections

-        What does “normal” mean?

- removed the word normal to reflect just the overview of the immunology of infection

-        This section is difficult to follow. A number of immune response effectors are listed and described, but it is difficult to gain an integrated picture. A suggestion is to reference a detailed review on the topic, and provide here a summary of key aspects of the response that contribute to containment, others that allow for Mtb to remain viable and latent, and finally the host factors that contribute to Mtb replication and disease. Indicate what is not known.

            -Thank you for this feedback, after reading the paragraph we saw how it was easy to get lost. Therefore, to help remedy this I removed some of the very specific detail and kept it more general of an overview of the immune system response. we added some more information on the topics you suggested and also added a table to show the different cells and their responses at different phases of the infection. Due to the extensive edits on this part of the manuscript, the changes were made on a separate document and noted in the manuscript as a newly edited pasted document for cleanliness sake. 

Autophagy, mitophagy:

-        Long paragraphs with lots of detail, but overall little relevance to Mtb in the elderly. Suggest that the focus is more towards the role of autophagy and mitophagy in Mtb containment, and then how this may be altered in old age.

            -Again, this was meant to be an overview of Autophagy and Mitophagy as it stands in “normal and healthy young” individuals. The aging aspect is covered later in the manuscript under the “How Ageing Affects mTOR Signaling, Autophagy & Mitophagy” section. This is to give an overview of what the normal princess looks like in containing M. tb infection and using the latter section to compare it to the elderly.

New formatting to the sections and subheadings make this clearer. Edits have been made in large part to this paragraph to simplify and make the writing easier to follow and comprehend. A supplemental table/figure has also been added to additionally supply simplified understanding of this very complex pathway highlighting the most important steps in a general overview. Two figures have been added to illustrate the processes of autophagy and mitophagy in physiological circumstances.

mTOR signaling:

-        Section with lots of details on pathways. Very difficult to follow, including its relationship to aging.

            -Due to the extensive edits on this part of the manuscript, the changes were made on a separate document and noted in the manuscript as a newly edited pasted document for cleanliness sake. Again, this was meant to be an overview of mTOR as it stands in “normal and healthy young” individuals. The aging aspect is covered later in the manuscript under the “How Ageing Affects mTOR Signaling, Autophagy & Mitophagy” section. New formatting to the sections and subheadings make this clearer.

Edits have been made in large part to this paragraph to simplify and make the writing easier to follow and comprehend. A supplemental table/figure has also been added to additionally supply simplified understanding of this very complex pathway highlighting the most important steps in a general overview.

  1. 8. Ageing Effects on Immune System Pathway

- I thought this section would describe the known defects in immune function in the elderly.  Instead, it focuses on mitochondrial dysfunction, how it leads to lower ATP, and how this broadly affects immune function and other processes. So the section title does not coincide with its content.

- Thank you for this comment. After re-evaluating, we agree with your suggestion and thus went ahead and changed the title of this section to “Ageing Effects on Mitochondrial Function” in order to better reflect the objective of this section of the manuscript.

  1. Ageing Within the T-Cell System

- 1st paragraph is difficult to understand

- Nice description of the mechanism by which T-cell aging may occur.  However, with respect to TB, the summary of results is only one-sided, and data interpretation is more complex.  Only the defects in CD4 T-cells are summarized, but data on the early resistance conferred by CD8 T cells is not described. All these studies are in mice. Furthermore, there is no summary of other studies where no defects were identified in T-cell mediated responses of humans in response to mycobacterial antigens.  See Ault, Dwivdi et al, 2018; Scordo, Aguillón-Durán et al, 2021.  Consolidation of the different findings is important to provide a broader understanding.

-We appreciate your feedback and have decided to add to the 1st paragraph to include how ageing affects CD8+ T-cells and what that means to the immune system. We also changed some sentences to make it easier to understand. Additionally, we added evidence from the articles you provided supporting “no difference in T-cell response with regard to age.” Although there may not be a difference in T-cell response to infection, the overall T-cell number decreases with age and therefore “collectively” the immune response capability to fight infection. 

  1. How Ageing Affects mTOR Signaling, Autophagy & Mitophagy

- line 440. Do you mean “mitochondria”, and not “cells”?

- The 2nd to last sentence infers that mTOR-induced autophagy should increase risk of Mtb in the elderly. Suggest the authors provide examples of what is known in younger hosts with respect to this hypothesis. This would strengthen the case.  It would be of more value if the first paragraphs of this section be consolidated, and this last component should be expanded.

- Figure 3. How may this affect TB susceptibility?  Also, this figure does not show autophagy, which is the topic of this section.

- We realize the sentence you are referring to in line 440 did not quite make sense. We went ahead and changed the sentence around to make more sense. We also were referring to autophagy in a broad sense and therefore meant “cells”. In the following paragraph, we specify mitophagy as well as bring in proof from a research study that there is in fact overactivation of mTORC1 pathway with regards to age. Therefore, younger hosts are not as inclined to build up misfolded proteins and organelle damage.

  1. Ageing Effect on GSH System

- Nice section

Thanks much.

  1. Genes/Transcription Factors that play a role in ageing

- Needs a figure

-Thank you for this insight and we agree, it needed a figure. To illustrate the important genes and transcription factors that respond to oxidative stress, we included a supplementary figure to summarize the signaling pathways involved in aging.

  1. Ageing effects on susceptibility to M. tb infection and possible preventative strategies:

- The recommendation for giving BCG vaccination is unclear. This topic is only brought up here and with brief foundation for its use. Of note, in Scordo et al, PlosOne 2021 report that no BCG vaccination at birth is associated with higher odds of TB.  However, these findings require further study before being considered as a basis for a recommendation for BCG in old age.  Data on BCG use and protection against COVID-19 has not supported its protective role for this viral infection.

 -Thank you for your input! We revised this section to state our current recommendation involving BCG vaccination as well as included information from Scordo et al, PlosOne 2021 and recommendations from the CDC. We have found based on our research that BCG vaccination does in fact have a protective effect and therefore should be considered. We also suggest that more research is needed to fully understand the benefits of the BCG vaccination as well as ways to improve its protective effects.

- With regards to BCG use and protection against COVID-19, we did not research nor discuss any potential use of the BCG vaccine against COVID-19 and are not planning to investigate this connection in this manuscript.

Table 1. 

Overall unclear how this table fits with title of the study.  Where is the connection between aging and TB?

What does “ATP-dependent immune function" mean?  All immune function needs ATP.

-Thank you for pointing this out. We changed the heading of the table to better reflect the title and focus of this paper. The purpose of this table is to summarize the major mechanisms of age-mediated immune dysfunction which leads to increased susceptibility to M. tb infection.

Figure 1.  T-cell cytoplasm seems disproportionate (too big)

-We reduced the size of the cytoplasm to reflect a more proportionate T-cell.

Reviewer 2 Report

The main focus of this review is on the effect of aging on the immediate system related to TB. The review is quite comprehensive regarding the general effects of aging. However, since the focus (according to the title)  is on M. tuberculosis and TB, a more thorough discussion related to reports showing that TB immunity is really affected by aging.

Minor points:

It is stated that 2.9 million deaths are related to TB. According to WHO, the number is 1.5 million.

autopiphagy should be autophagy

Author Response

Thank you for all of your feedback, comments, kudos and suggestions. The manuscript was extensively revised to incorporate all the recommendations and suggestions of the reviewers. The edits made are denoted as track changes. We hope to have improved this paper to your satisfaction and hope we addressed all the comments and concerns in a timely and knowledgeable manner. We look forward to the publishing this important work. Awaiting to hear back from you regarding the numerous changes that were made.

The main focus of this review is on the effect of aging on the immediate system related to TB. The review is quite comprehensive regarding the general effects of aging. However, since the focus (according to the title)  is on M. tuberculosis and TB, a more thorough discussion related to reports showing that TB immunity is really affected by aging.

Appreciate your comment. In the revised manuscript we included a very thorough discussion on how immunity to Mtb is affected by ageing.

Minor points:

It is stated that 2.9 million deaths are related to TB. According to WHO, the number is 1.5 million

  • Fixed this data point

autopiphagy should be autophagy

  • Not sure where this was in the manuscript, we couldn’t find where this error was

Reviewer 3 Report

The title of the paper was kind of misleading as mostly the paper talks about the immunological facts that could be implemented for any other diseases not just for TB. It will be great if the review was focused on certain topics rather than being general and repeating the same information several times. It is also important that authors should use recent references.

Author Response

Thank you for all of your feedback, comments, kudos and suggestions. The manuscript was extensively revised to incorporate all the recommendations and suggestions of the reviewers. The edits made are denoted as track changes. We hope to have improved this paper to your satisfaction and hope we addressed all the comments and concerns in a timely and knowledgeable manner. We look forward to the publishing this important work. Awaiting to hear back from you regarding the numerous changes that were made.

The main focus of this review is on the effect of aging on the immediate system related to TB. The review is quite comprehensive regarding the general effects of aging. However, since the focus (according to the title)  is on M. tuberculosis and TB, a more thorough discussion related to reports showing that TB immunity is really affected by aging.

We substantially revised the manuscript by including comprehensive information on how TB immunity is affected by aging. To further enhance the clarity of the complex mechanisms that detailed in the manuscript we included additional figures and table. The revised manuscript has six figures and two tables.

Minor points:

It is stated that 2.9 million deaths are related to TB. According to WHO, the number is 1.5 million.

Thanks for pointing this. We made changes

autopiphagy should be autophagy

This has been changed
